# Education and income-related inequalities in multimorbidity among older Brazilian adults

**Fabíola Bof de Andrade**[1]*, **Elaine Thumé**[2], **Luiz Augusto Facchini**[3], **Juliana Lustosa Torres**[4], **Bruno Pereira Nunes**[5]

**1** René Rachou Institute, Oswaldo Cruz Foundation (FIOCRUZ), Belo Horizonte, MG, Brazil, **2** Collective Health Nursing Department, Federal University of Pelotas, Pelotas, RS, Brazil, **3** Social Medicine Department, Federal University of Pelotas, Pelotas, RS, Brazil, **4** Preventive and Social Medicine Department, Federal University of Minas Gerais, Belo Horizonte, MG, Brazil, **5** Department of Nursing, Federal University of Pelotas, Pelotas, RS, Brazil

* fabiola.bof@fiocruz.br

## Abstract

### Objectives

This study aimed to evaluate the existence of socioeconomic inequalities related to the prevalence of multimorbidity in the Brazilian population aged 60 and older.

### Methods

This was a cross-sectional study with data from the last Brazilian National Health Survey (PNS) collected in 2019. Multimorbidity was the dependent variable and was defined as the presence of two or more chronic diseases. All the diseases were assessed based on a self-reported previous medical diagnosis. Education and per capita family income were the measures of socioeconomic position. Socioeconomic inequalities related to multimorbidity were assessed using two complex measures of inequality; the Slope Index of Inequality (SII) and the Concentration Index (CI).

### Results

The prevalence of multimorbidity in Brazil was 56.5% 95% CI (55.4; 57.6) and varied from 46.9% (44.3; 49.6) in the North region to 59.3% (57.0; 61.5) in the South region. In general, individuals with higher socioeconomic positions had a lower prevalence of multimorbidity. Significant absolute and relative income inequalities were observed in the South region [SII -9.0; CI -0.054], Southeast [SII -9.8; CI -0.06], and Middle-east [SII -10.4; CI -0.063]. Absolute and relative education inequalities were significant for the country and two of its regions (Southeast [SII -12.7; CI -0.079] and South [SII -19.0; CI -0.109]).

### Conclusions

The prevalence of multimorbidity is high in Brazil and all of its macro-regions. The significant findings concerning the inequalities suggest that the distribution of this condition is more concentrated among those with lower education and income.

**Data Availability Statement:** The data underlying the results presented in the study are available from https://www.ibge.gov.br/estatisticas/sociais/

saude/9160-pesquisa-nacional-de-saude.html?
=&t=microdados.

**Funding:** FBA - This work was supported by
Foundation for Research Support from Minas
Gerais State (FAPEMIG), Brazil [grant APQ-00573-
21]. http://www.fapemig.br/pt/ Funder role: none.

**Competing interests:** The authors have declared
that no competing interests exist.

## Introduction

Chronic non-communicable diseases have been declared a global epidemic since 2011 [1]. It is
estimated that more than 50% of the elderly population has two or more chronic diseases, a
condition known as multimorbidity [2, 3]. In Brazil, approximately a quarter (24.2%) of the
population aged 18 years and over has two or more chronic diseases [4] and for the population
aged 60 years and over, the prevalence is 53.1% [5].

The accelerated population aging, especially in developing countries, makes multimorbidity
one of the greatest challenges for health services, due to its association with functional decline
[6, 7], use of health care services [8, 9] and mortality [10–12]. Among the different determi-
nants of this condition, the impact of socioeconomic conditions has been increasingly studied
with mixed results, depending on the measure of socioeconomic position and the country's
economic level [13–22].

A meta-analysis with 10 cross-sectional studies, representing 13 different populations,
found that individuals with lower educational levels were 64% more likely to have multimor-
bidity than the ones with higher education. However, there was substantial heterogeneity
between studies partly explained by the method of multimorbidity ascertainment. Moreover,
different studies exploring measures of deprivation found a consistent positive association,
while the evidence on income was mixed [13]. Studies addressing the socioeconomic inequali-
ties related to multimorbidity in the elderly population are less common. In Brazil, among
adults [4] and older adults [22, 23], they point to a negative association with schooling [4, 22,
23] and a positive association with the asset index [23]. In addition, few studies worldwide
have evaluated these inequalities through complex measures, which are important to assess the
effects of social policy [24].

The continuous evaluation of health inequalities is essential for reducing health disparities
and its monitoring is an important target to achieve the United Nations Sustainable Develop-
ment Goals [25]. Thus, this study aimed to evaluate the existence of socioeconomic inequalities
related to the prevalence of multimorbidity in the Brazilian population aged 60 and older.

## Methods

This was a cross-sectional study with data from the last Brazilian National Health Survey
(PNS) collected in 2019. This is a household survey carried out by the Brazilian Institute of
Geography and Statistics (*Instituto Brasileiro de Geografia e Estatística*–IBGE) in partnership
with the Ministry of Health.

The survey sample is representative of the Brazilian population residing in permanent pri-
vate households, the five-country macro-regions, federation units, capitals, metropolitan areas
and urban and rural regions. Sample selection was made by clusters into three stages. The pri-
mary sample unit was the census tracts or sets of tracts. In the second stage households from
the National Register of Addresses for Statistical Purposes were selected. In the third stage, one
resident aged 15 years old or over from each household was randomly selected. [26]

Informed consent was obtained directly on the mobile data collection devices before the
interview [26]. The survey data collections were approved by the National Commission for
Ethics in Research/National Health Council, under protocol number 3,529,376. [26]

The present study included participants aged 60 and older with complete information for
the variables of interest including a sample of 22,728 individuals.

### Dependent variable

Multimorbidity was defined as the presence of two or more chronic diseases, as proposed by
the World Health Organization [27]. All the diseases were self-reported and included the

following conditions: high blood pressure, diabetes, high cholesterol, heart disease, stroke, asthma, arthritis or rheumatism, chronic back problem, chronic obstructive pulmonary disease, work-related musculoskeletal disorder, cancer, chronic kidney failure, depression, other mental illness. Each of the diseases was evaluated through the following question, except for chronic back problem and other mental diseases: "*Has a doctor ever diagnosed you with [name of disease]*?". chronic back problem, depression and other mental health diseases were evaluated by the questions, respectively: "*Do you have any chronic back problems, such as chronic back or neck pain, low back pain, sciatica, spinal or disc problems*?"; "*Has a doctor or mental health professional (such as a psychiatrist or psychologist) ever given you a diagnosis of depression*?" and "*Has a doctor or healthcare professional (such as a psychiatrist or psychologist) ever given you the diagnosis of another mental illness, such as an anxiety disorder, from panic, schizophrenia, bipolar disorder, psychosis or OCD (Obsessive Compulsive Disorder) etc*.?"

### Independent variables

Measures of socioeconomic position (i.e., education and per capita family income) were the independent variables of interest.

Education was categorized into six categories (no education, incomplete elementary school, complete elementary school, incomplete secondary, complete secondary, some college or more) and per capita family income into five categories (up to 1 minimum wage, >1–2 minimum wages, >2–3 minimum wages, >3–5 minimum wages, >5 minimum wages).

### Data analyses

Data analyses included the description of the sample and the distribution of the multimorbidity according to the measures of socioeconomic position. Socioeconomic inequalities related to multimorbidity were assessed using two complex measures of inequality; the Slope Index of Inequality (SII) and the Concentration Index (CI).

The SII is an absolute regression-based measure of inequality that takes the entire socioeconomic distribution into account, rather than just comparing the two most extreme groups [28]. To calculate the SII each category of the socioeconomic position measure (i.e. education, income) is assigned a relative position score based on the midpoint of the range of the cumulative distribution of the population of participants in each category. Individuals were cumulatively ranked from 0 to 1 according to socioeconomic position, in ascendent order, such that "0" represents the lowest level of socioeconomic position and "1" represents the highest level of socioeconomic position. The relative position variable will then be entered as an independent variable in the regression model. Because the outcome is binary, the SII was estimated with logistic regression to avoid predicting values in the regression model that were outside the interval between 0 and 1. [29] The SII is the difference in probability (absolute inequality) between those at the highest level of socioeconomic status and those at the lowest level. An SII value lower than zero indicated that the prevalence of multimorbidity is higher among populations in lower socioeconomic positions [24].

The CI proposed by Wagstaff was used in the study to analyze the relative inequalities [30]. This measure can be used to show how the health indicator is distributed across individuals that are ranked by a socioeconomic position measure. The concentration index is defined as twice the area between the concentration curve and the line of equality [24, 30]. CI varies between -1 and +1; when there is no socioeconomic-related inequality, CI is zero. A positive value indicates that the rich have a higher prevalence of multimorbidity than the poor (pro-rich inequality), and a negative value indicates a disproportionate concentration of multimorbidity among poor individuals (pro-poor). [24] The CI was estimated using the Stata

command conindex. Details about its estimation have been reported previously [30]. All analyses were performed using the Stata 15.0 software and took into account the complexity of the survey sampling design.

## Results

Table 1 shows the characteristic of the Brazilian older adults included in the study. Most of the participants were female and had 60–69 years old. Considering the socioeconomic position measures, a higher proportion of individuals had no education (16.8%) or incomplete elementary school (46.5%). About 41.7% reported receiving up to one minimum wage, whereas 7.4% reported 5 or more.

The prevalence of multimorbidity in Brazil was 56.5% 95% CI (55.4; 57.6) and varied from 46.9% (44.3; 49.6) in the North region to 59.3% (57.0; 61.5) in the South region. The prevalence in the Northeast, Middle-east and Southeast were, respectively, 53.1% (51.2; 55.0), 57.3% (54.6; 60.0) and 58.5% (56.6; 60.4).

The distribution of multimorbidity by education and family income per capita is presented in Fig 1. In general, individuals with higher socioeconomic positions had a lower prevalence of multimorbidity, but different patterns were found and a linear pattern was only observed for the southeast region concerning income.

**Table 1. Distribution of the population.**

|  | % (95% CI) |
|---|---|
| **Age** (years) | |
| 60–69 | 56.3 (55.2–57.4) |
| 70–79 | 30.1 (29.2–31.1) |
| 80+ | 13.6 (12.8–14.3) |
| **Sex** | |
| Male | 43.3 (42.2–44.4) |
| Female | 56.7 (55.6–57.8) |
| **Income** | |
| Up to 1 minimum wage | 41.7 (40.6–42.9) |
| >1–2 minimum wages | 31.9 (30.8–32.9) |
| >2–3 minimum wages | 10.8 (10.1–11.5) |
| >3–5 minimum wages | 8.2 (7.6–8.9) |
| >5 minimum wages | 7.4 (6.7–8.1) |
| **Education** | |
| No education | 16.8 (16.0–17.7) |
| Incomplete elementary school | 46.5 (45.3–47.6) |
| Complete elementary school | 6.8 (6.3–7.4) |
| Incomplete secondary | 2.7 (2.4–3.1) |
| Complete secondary | 14.5 (13.7–15.4) |
| Some college or more | 12.7 (11.9–13.6) |
| **Country regions** | |
| North | 6.1 (5.8–6.4) |
| Northeast | 25.4 (24.5–26.2) |
| Southeast | 46.4 (45.3–47.6) |
| South | 15.7 (15.1–16.4) |
| Middle-east | 6.4 (6.0–6.8) |

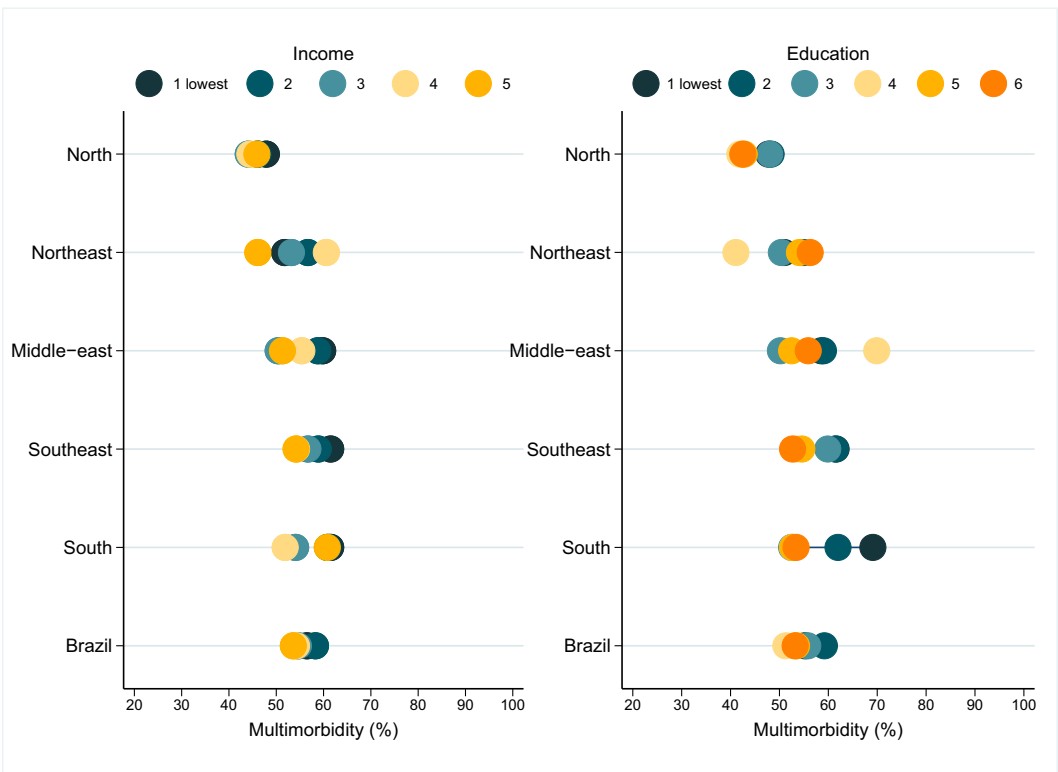

**Fig 1. Prevalence of multimorbidity by income and education.**

Significant absolute and relative income inequalities were observed in the South region [SII -9.0; CI -0.054], Southeast [SII -9.8; CI -0.06] and Middle-east [SII -10.4; CI -0.063], being the prevalence of multimorbidity more concentrated among individuals with lower socioeconomic positions. Education inequalities were in the same direction as income inequalities. Absolute and relative indexes were significant for the country and two of its regions (Southeast [SII -12.7; CI -0.079] and South [SII -19.0; CI -0.109]). In the country, the prevalence of multimorbidity was 5.1 percent points higher among individuals at the lower educational level than in those at the higher and the concentration index was -0.031 (Table 2).

**Table 2. Slope index and concentration index of inequality by education and income.**

|  | Income | Education | Income | Education |
|---|---|---|---|---|
|  | SII (95% CI) | SII (95% CI) | CIX (95% CI) | CIX (95% CI) |
| **Brazil** | -2.7 | -5.1 | -0.015 | -0.031 |
|  | (-6.9; 1.5) | (-8.9; -1.3) | (-0.039; 0.010) | (-0.054; -0.008) |
| **North** | -5.3 | -6.0 | -0.025 | -0.036 |
|  | (-15.6; 5.1) | (-15.0; 3.0) | (-0.074; 0.024) | (-0.089; 0.018) |
| **Northeast** | 6.2 | 4.5 | 0.027 | 0.027 |
|  | (-2.4; 14.7) | (-1.9; 10.9) | (-0.009; 0.064) | (-0.012; 0.065) |
| **Southeast** | -9.8 | -12.7 | -0.060 | -0.079 |
|  | (-16.7; -2.8) | (-19.2; -6.2) | (-0.103; -0.017) | (-0.120; -0.038) |
| **South** | -9.0 | -19.0 | -0.054 | -0.109 |
|  | (-17.7; -0.3) | (-26.6; -11.3) | (-0.107; -0.002) | (-0.153; -0.065) |
| **Middle-east** | -10.4 | -6.2 | -0.063 | -0.038 |
|  | (-20.4; -0.4) | (-16.7; 4.2) | (-0.124; -0.002) | (-0.102; 0.026) |

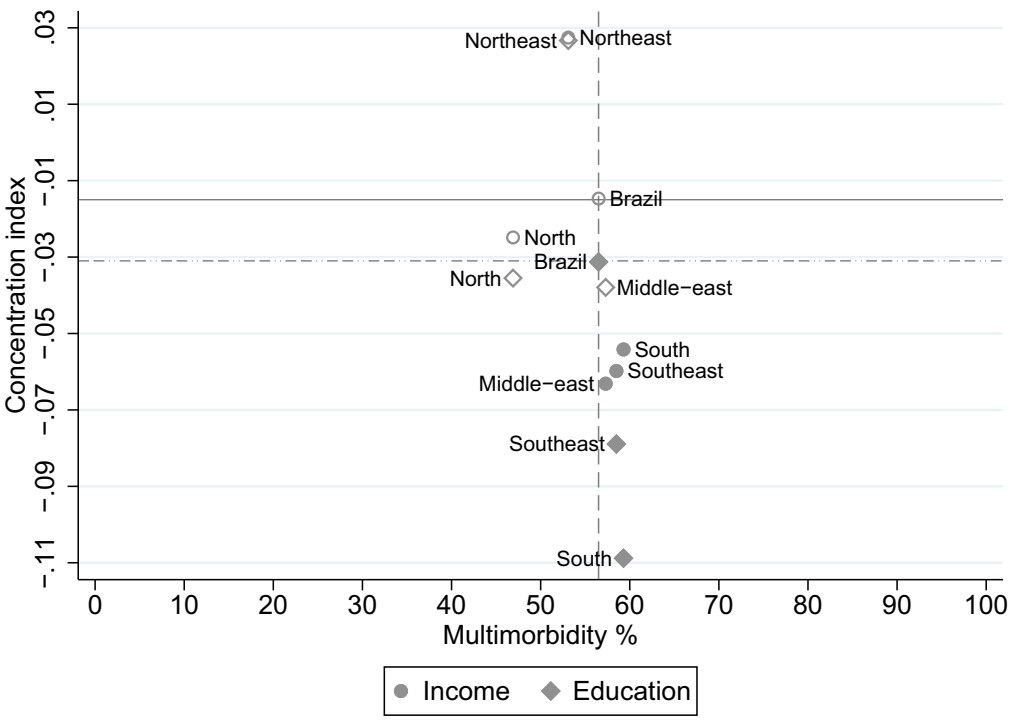

**Fig 2. Concentration index by the prevalence of multimorbidity.** Notes for Fig 2: Hollow markers are non-significant. Horizontal dashed line = education CI. Horizontal solid line = income CI.

The quadrant graph (Fig 2) was used to compare the CI and prevalence of multimorbidity for each region with the corresponding values of the country. Most of the regions (Middle-east, South and Southeast), for income inequalities, were concentrated in the bottom right quadrant, but the magnitude of the CI was similar among these regions.

## Discussion

This study showed a high prevalence of multimorbidity in Brazil and all of its macro-regions. The findings suggest that older adults in lower socioeconomic positions have a higher prevalence of multimorbidity. An absolute and relative education inequality was observed within the country, whereas significant inequalities for both education and income were observed within the Southeast and South regions. Within the Middle-east region, only income-related inequalities were observed.

Previous studies using either the SII or CI to evaluate inequalities in multimorbidity used a different combination of diseases and a variety of socioeconomic position measures [i.e. asset index [21, 31], income [32], purchasing power [33] and schooling [33, 34]] making it difficult a direct comparison between the results. In most of the countries evaluated in these studies [21, 31, 32, 34], the concentration index was negative as in the present study, thereby demonstrating a pro-poor inequality, where the prevalence of multimorbidity is highly concentrated among the poor. None of the studies exclusively investigated older adults. The study by Kunna et al. evaluated a sample of individuals 50 and older from China and Ghana. Within the latter country, the CI for household assed-inequalities was pro-poor CI -0.0801 (95% CI -0.1233;

-0.0368), but in the former country it was found to be pro-rich [CI 0.1453 (95% CI 0.0794; 0.2083)], being the prevalence of multimorbidity more concentrated among the rich [21].

In Brazil, two previous studies used both of these measures of inequalities and found opposite results [33, 34]. Findings from a nationwide sample of Brazilian adults, aged 20 to 59 years [33] reported pro-rich absolute and relative inequality by purchasing power and education among adult men. On the other hand, inequalities were absent among women. Unlike the latter study and corroborating the present findings, Delpino et al. [34] found that, for both absolute and relative inequalities, multimorbidity was more concentrated among Brazilian adults with lower educational level in 2013 SII -11.2 (IC95% -13.6; -8.7); CI -0.07 (95% CI -0.09; -0.05) and in 2019 [SII -10.1 (IC95% -12.2; -7.9)].; CI -0.05 (95% CI -0.07; -0.04).

The magnitude of absolute inequalities was higher than the relative ones and the findings also suggest that they are smaller than the ones observed among Brazilian adults in the same year [34]. These differences may rely on the fact that the prevalence of multimorbidity is higher among older adults which has a direct impact on the estimated indexes. In contexts where the prevalence is high, it is expected a smaller relative inequality in comparison to absolute. The concentration index tends to be lower at larger overall levels of the outcome [35] and absolute measures are expected to be low at both very low and very high overall levels of the outcome [36]. Moreover, these measures might give different answers to the existence of inequalities, although in the present study similar findings were confirmed by both. Thus, besides reporting both measures of absolute and relative inequalities, the prevalence of the outcome should be presented alongside inequality data to give a better understanding of the scenario [24].

As observed in the country, the prevalence of multimorbidity was high in its regions, which showed similar patterns for income and education inequalities. Considering the magnitude of the indexes for the country and its regions, on the contrary of what was observed for absolute inequalities, relative indexes might be considered reasonably small, although there is no consensus on a specific cutoff point. Accordingly, it has been reported [24] that absolute values for the concentration index rarely exceed 0.5, then values of 0.2 to 0.3 are considered to represent a reasonably high level of relative inequality.

This study built on the existing literature by estimating for the first time the magnitude of relative and absolute inequalities, among older adults, using complex measures and two socioeconomic position stratifiers (i.e., schooling and household per capita income). The findings show that the direction of the inequalities was the same for both income and education, but a different pattern of significance was found. As for the relative and absolute indexes, different socioeconomic position measures might give different results as the mechanisms by which they impact health throughout life are distinct, although they may overlap [37, 38]. Income directly accesses actual material resources and might represent health-relevant disposable income for the individuals [37] (e.g., access to medical diagnosis). Moreover, at retirement ages income are more stable among individuals [39]. Education, on the other hand, captures the long-term influences of early life circumstances on adult health and the influence of adult resources on health as it is also related to income and wealth in the life course [40]. Another strength of this measure is its relation to health literacy, thereby influencing the individual's ability to understand, report and engage with health information and services [40, 41] and to adopt healthier behavior.

The major strength of this study is the use of a national sample of older adults and the possibility to evaluate the existence of inequalities in Brazil and its macro-regions, which are historically different. The Brazilian National Health Survey, carried out in 2019, is the most recent household health survey in the country, due to its systematic methodology and periodicity, is appropriate for this analysis and makes it possible the comparison of its findings throughout the time. The use of complex measures of inequalities is also worth noting. Among the

limitations, the restricted number of diseases and as well as the use of self-reported diseases might have impacted the prevalence of multimorbidity towards a lower estimate. However, the use of a question addressing a physician diagnosis might improve the report [42, 43], albeit a small effect is expected among individuals in the lower socioeconomic position for which the access to medical diagnosis is least frequent.

Based on the findings of this study, it is possible to conclude that multimorbidity affects a significant proportion of the older Brazilian population and its prevalence is high in all of the country's macro-regions. The higher prevalence of this condition is more concentrated among the ones with lower education and income in the country. There are differences in the distribution of relative and absolute inequalities among the country regions. These findings highlight the importance of continually evaluating within regions' differences as the country's mean values of inequalities hide the specificities of each of them.

## Author Contributions

**Conceptualization:** Fabíola Bof de Andrade, Elaine Thumé, Luiz Augusto Facchini, Juliana Lustosa Torres, Bruno Pereira Nunes.

**Formal analysis:** Fabíola Bof de Andrade.

**Funding acquisition:** Fabíola Bof de Andrade.

**Methodology:** Fabíola Bof de Andrade.

**Visualization:** Fabíola Bof de Andrade.

**Writing – original draft:** Fabíola Bof de Andrade.

**Writing – review & editing:** Fabíola Bof de Andrade, Elaine Thumé, Luiz Augusto Facchini, Juliana Lustosa Torres, Bruno Pereira Nunes.

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
