## [Decision Letter · Decision Letter 0]

4 Aug 2022

PONE-D-22-16455Education and income-related inequalities in multimorbidity among older Brazilian adultsPLOS ONE

Dear Dr. Bof de Andrade,

Thank you for submitting your manuscript to PLOS ONE. After careful consideration, we feel that it has merit but does not fully meet PLOS ONE’s publication criteria as it currently stands. Therefore, we invite you to submit a revised version of the manuscript that addresses the points raised during the review process.

ACADEMIC EDITOR:Please consider all suggestions made by reviewers.==============================

We look forward to receiving your revised manuscript.

Kind regards,

Marcus Tolentino Silva

Academic Editor

PLOS ONE

Journal Requirements:

FBA - This work was supported by Foundation for Research Support from Minas Gerais State (FAPEMIG), Brazil [grant APQ-00573-21]. 

http://www.fapemig.br/pt/

Funder role: none.  

Reviewers' comments:

Reviewer's Responses to Questions

**Comments to the Author**

1. Is the manuscript technically sound, and do the data support the conclusions?

Reviewer #1: Yes

Reviewer #2: Yes

2. Has the statistical analysis been performed appropriately and rigorously? 

Reviewer #1: Yes

Reviewer #2: Yes

3. Have the authors made all data underlying the findings in their manuscript fully available?

Reviewer #1: Yes

Reviewer #2: Yes

4. Is the manuscript presented in an intelligible fashion and written in standard English?

Reviewer #1: Yes

Reviewer #2: Yes

5. Review Comments to the Author

Reviewer #1: Bof de Andrade and colleagues conducted a cross-sectional analysis to explore the association between socioeconomic inequalities and the prevalence of multimorbidity in the general population aged 60 years or older, using nationally-representative data retrieved from the most recent Brazilian National Health Survey. The study findings suggest that individuals with higher socioeconomic status tend to have a lower prevalence of multimorbidity in Brazil.

The study carries important implications for health equity across different regions, despite room for improvement through a minor revision.

The estimation of the slope index of inequality (SII) and the concentration index (CI), which were adopted as measures of inequality in the study, were not explicitly explained. For example, it remains to be elucidated as to how the regression models were constructed to yield the inequality coefficient. Besides, the authors may wish to add a bit more elaboration on how the confounding factors were accounted for in the study.

Reviewer #2: The lines in the article are not numbered.

In the abstract the conclusion should provide more information on the distribution of multimorbidity in the country, as it is, the conclusion is to general and more detailed information on the most affected areas of the country could be helpful.

In the first paragraph of the introduction reference 4 is in () and not in [].

Methods: In the independent variables, the education variable is defined with six categories, but one category is missing in the parenthesis, but then it is mentioned in table 1. Add the category in the text “some college or more”.

In the discussion, a closing paragraph should be included regarding the inequality differences found in the different regions of the country and its relationship with the high multimorbidity prevalence. This was not evident in the discussion.

There is no conclusion in the manuscript, include it

6. PLOS authors have the option to publish the peer review history of their article (what does this mean?). If published, this will include your full peer review and any attached files.

Reviewer #1: **Yes: **Harry H.X. Wang

Reviewer #2: No

---

## [Author Response · Author response to Decision Letter 0]

11 Aug 2022

Reviewer #1: 

Bof de Andrade and colleagues conducted a cross-sectional analysis to explore the association between socioeconomic inequalities and the prevalence of multimorbidity in the general population aged 60 years or older, using nationally-representative data retrieved from the most recent Brazilian National Health Survey. The study findings suggest that individuals with higher socioeconomic status tend to have a lower prevalence of multimorbidity in Brazil.

The study carries important implications for health equity across different regions, despite room for improvement through a minor revision.

The estimation of the slope index of inequality (SII) and the concentration index (CI), which were adopted as measures of inequality in the study, were not explicitly explained. For example, it remains to be elucidated as to how the regression models were constructed to yield the inequality coefficient. Besides, the authors may wish to add a bit more elaboration on how the confounding factors were accounted for in the study.

Response: Thanks for the careful review of our manuscript and for the suggestions. We have included more information on how to estimate the indexes and included references for the papers that detailed explained the methods. Regarding the confounders, they were not discussed because in these analyses only the outcome and the socioeconomic position measures are included. 

Reviewer #2: 

The lines in the article are not numbered.

Response: Thanks for the careful review of our manuscript and for the suggestions. We have included the lines numbers in the paper.

In the abstract the conclusion should provide more information on the distribution of multimorbidity in the country, as it is, the conclusion is to general and more detailed information on the most affected areas of the country could be helpful.

Response: Thanks for the suggestion. We included this information in the abstract. 

In the first paragraph of the introduction reference 4 is in () and not in [].

Response: We have made the change.

Methods: In the independent variables, the education variable is defined with six categories, but one category is missing in the parenthesis, but then it is mentioned in table 1. Add the category in the text “some college or more”.

Response: We have included the category.

In the discussion, a closing paragraph should be included regarding the inequality differences found in the different regions of the country and its relationship with the high multimorbidity prevalence. This was not evident in the discussion.

Response: We highlighted this information in the discussion section in the paragraph where the magnitude of the inequalities is discussed.

There is no conclusion in the manuscript, include it

Response: We have reviewed the discussion to better address the conclusions.

---

## [Decision Letter · Decision Letter 1]

26 Sep 2022

Education and income-related inequalities in multimorbidity among older Brazilian adults

PONE-D-22-16455R1

Dear Dr. Bof de Andrade,

We’re pleased to inform you that your manuscript has been judged scientifically suitable for publication and will be formally accepted for publication once it meets all outstanding technical requirements.

Kind regards,

Marcus Tolentino Silva

Academic Editor

PLOS ONE

Additional Editor Comments (optional):

Reviewers' comments:

Reviewer's Responses to Questions

**Comments to the Author**

1. If the authors have adequately addressed your comments raised in a previous round of review and you feel that this manuscript is now acceptable for publication, you may indicate that here to bypass the “Comments to the Author” section, enter your conflict of interest statement in the “Confidential to Editor” section, and submit your "Accept" recommendation.

Reviewer #1: All comments have been addressed

Reviewer #2: All comments have been addressed

2. Is the manuscript technically sound, and do the data support the conclusions?

Reviewer #1: Yes

Reviewer #2: Yes

3. Has the statistical analysis been performed appropriately and rigorously? 

Reviewer #1: Yes

Reviewer #2: Yes

4. Have the authors made all data underlying the findings in their manuscript fully available?

Reviewer #1: Yes

Reviewer #2: Yes

5. Is the manuscript presented in an intelligible fashion and written in standard English?

Reviewer #1: Yes

Reviewer #2: Yes

6. Review Comments to the Author

Reviewer #1: Thank the authors for addressing my previous comments. The paper can be considered for acceptance for publication.

Reviewer #2: One last revision: in line 287, the redaction is confusing, change “these findings the importance of…” to “these findings highlight the importance of”

7. PLOS authors have the option to publish the peer review history of their article (what does this mean?). If published, this will include your full peer review and any attached files.

Reviewer #1: **Yes: **Harry H.X. Wang

Reviewer #2: No

---

## [Editor Report · Acceptance letter]

3 Oct 2022

PONE-D-22-16455R1 

Education and income-related inequalities in multimorbidity among older Brazilian adults  

Dear Dr. Bof de Andrade:

I'm pleased to inform you that your manuscript has been deemed suitable for publication in PLOS ONE. Congratulations! Your manuscript is now with our production department. 

Kind regards, 

on behalf of

Dr. Marcus Tolentino Silva 

Academic Editor

PLOS ONE